# F-PvE: Fairness-Aware Structured Neural Network Pruning via Evolution

## Abstract

Model compression plays a crucial role in deployment, as it can significantly reduce computational costs with minimal loss in accuracy. However, recent studies have shown that model compression may involve additional bias, posing fairness risks that can potentially lead to social impact. As a result, mitigating bias during model compression has emerged as an important topic. In this work, we focus on structured neural network pruning, a widely adopted model compression technique that remains rarely explored in the context of fairness. Specifically, we introduce evolutionary algorithms as a general yet powerful approach to achieve fairness-aware structured pruning. That is, we formulate structured pruning as a subset selection problem and use evolutionary search to identify an optimal set of structural components to retain, balancing both accuracy and fairness objectives. Given the multi-objective nature and the large combinatorial search space of structural components, we further incorporate multi-objective evolution and cooperative coevolution to effectively address them. To verify the effectiveness of our method, we conduct experiments that cover three typical fairness scenarios: class-wise and group-wise fairness in classification models, and toxicity in language models. Compared with classic structured pruning methods and state-of-the-art competitors on fairness-aware structured pruning, our method can preserve better fairness while keeping competitive accuracy, demonstrating the superiority of evolutionary optimization for fairness-aware structured pruning in practice.

## 1 Introduction

Recent years have witnessed the blossom of model ability and their applications. Nevertheless, the impressive performance usually come at the cost of increasing computational requirements, posing great challenges for deployment, especially for resource-limited scenarios (Menghani, 2023). To mitigate the burdens, model compression is playing an increasingly important role in practice, which can significantly reduce the computation cost from perspectives of both storage and inference time while keeping the original precision. In fact, prevailing model compression techniques including quantization (Rastegari et al., 2016), knowledge distillation (Hinton et al., 2015), and neural network pruning (Li et al., 2017) have been widely studied in the last decade. However, recent works (Hooker et al., 2019; 2020; Iofinova et al., 2023; Joseph et al., 2020; Stoychev & Gunes, 2022) indicate that the fairness of compressed models can be severely damaged, raising new concerns on the employment of model compression.

The fairness of machine learning models has been studied for a long history (Mehrabi et al., 2022), which is important especially in high-stack scenarios like diagnosis (Lin et al., 2023) and credit estimation (Shumovskaia et al., 2020). Generally, fairness measures the discrepancy of model performance on different classes or groups. For example, the models for hiring decisions should keep consistent precision on different social groups, preventing from potential bias (Cohen et al., 2020). Models for multi-class classification are expected to hold the same accuracy on each alternative class, avoiding the loss of efficacy on specific classes behind the overall good accuracy (Hooker et al., 2019). Meanwhile, the development of language models has also drawn significant attention to their fairness, particularly regarding toxicity words and social bias in generated content (Dhamala et al., 2021; Dixon et al., 2018). Accordingly, various metrics have been proposed to quantify the fairness of models from different views (Han et al., 2024). Great efforts have also been paid on improving fairness, such as data resampling (Yu, 2021), adversarial training (Madras et al., 2018) and

correction (Menon & Williamson, 2018), which can be referred in recent surveys (Mehrabi et al., 2022; Hort et al., 2023; Pessach & Shmueli, 2022).

Recently, a variety of works reveal that prevailing model compression methods can cause unexpected fairness degradation in resulted models. That is, compared with the original models, even that the overall performance of the compressed models is still good, the fairness of them can be explicitly damaged. (Hooker et al., 2019) first indicated that in compressed multi-class classification models, certain classes may suffer a disproportionately increasing portion of error, despite ostensibly good average performance of all classes. The following work (Hooker et al., 2020) demonstrates that certain groups of data bear disproportionately high portion of the error after compression. Paganini (2020) further illustrated the fairness issues and advocated a comprehensive evaluation of compressed models from a Pareto-optimal perspective, considering compression ratio, accuracy, and fairness simultaneously. These findings draw attention to the importance of compressing models in a fairness-aware manner, which has become an emerging topic.

Since various model compression methods are designed and implemented from different perspectives, it is impractical to simply enhance their fairness using a unified approach. Instead, previous studies typically focus on a single model compression method, such as knowledge distillation (Blakeney et al., 2021), quantization (Liu et al., 2025), and unstructured neural network pruning (Lin et al., 2022; Tang et al., 2023; Zhang et al., 2023), leaving structured neural network pruning rarely explored. The only prior work (Zayed et al., 2024) focuses on pruning the attention heads of Transformer-based language models using a greedy strategy, which exhaustively evaluates the impact of removing each head on the quality and fairness of generated contents, and retains the most effective ones. Nevertheless, it only considers a specific fairness scenario of mitigating toxicity in language models, and the greedy strategy is not scalable, which means that it is impractical for other common architectures such as Convolution Neural Networks (CNNs), where thousands of structural components are candidates for pruning. On the other side, although numerous works have explored fairness-aware unstructured neural network pruning, these approaches are difficult to be adapted to structured neural network pruning due to the discrete solution space of structural components. Moreover, structured neural network pruning is more valuable (He & Xiao, 2024), as the resulting regular structures are better suited to modern hardware and software, bringing genuine acceleration.

In this work, we focus on addressing fairness-aware structured neural network pruning. Prevailing structured neural network pruning methods are usually based on criteria like weight magnitude (Li et al., 2017). However, these criteria are not directly relevant to fairness, potentially leading to undesirable fairness in compressed models (Zayed et al., 2024). To enable fairness-aware structured neural network pruning in a wide range of scenarios, we propose a unified framework: Fairness-aware Pruning via Evolution (F-PvE). Specifically, we leverage evolutionary optimization, which has proven effective in related tasks such as conventional structured pruning (Zhou et al., 2021; Shang et al., 2022) and neural architecture search (Liu et al., 2023), to tackle the challenges of fairness-aware structured pruning. To this end, we formulate structured pruning as a subset selection problem, where the goal is to identify a subset of structural components to retain while pruning the rest. The evolutionary algorithm is designed to simultaneously optimize both accuracy and fairness of the resulting pruned network, which is fully determined by the selected subset of components. Noting that it involves multiple objectives and large-scale combinatorial search space of candidate components, we further employ multi-objective evolution (Zhou et al., 2011) and cooperative co-evolution (Ma et al., 2019) to address these challenges separately. That is, for architectures with relatively few components (e.g., pruning heads in Transformers), we adopt multi-objective evolution techniques such as NSGA-II (Deb et al., 2002) to achieve a better trade-off of accuracy and fairness. For architectures with a large number of components (e.g., CNNs), we turn to adopt cooperative coevolution, leveraging a divide-and-conquer strategy to enhance scalability and effectiveness.

To validate the effectiveness of our method in fairness-aware structured pruning, we conduct experiments across three representative fairness scenarios: class-wise degradation in multi-class classification, group-wise degradation in classification with sensitive attributes, and social bias in language models. These experiments cover both CNN and Transformer architectures. Experimental results show that our method consistently outperforms both prevailing structured pruning approaches and state-of-the-art fairness-aware structured pruning methods across all three tasks. Our main contributions are summarized as follows:

- We propose F-PvE, the first general approach for fairness-aware structured neural network pruning, which adopts evolutionary optimization techniques to identify fair subsets of structural units to retain in pruned models.

- To tackle the challenges arising from the multi-objective nature and the large combinatorial search space, we further incorporate multi-objective evolutionary algorithms and cooperative coevolution to enhance the performance.

- We conduct experiments across three representative fairness scenarios, covering the structured pruning of both CNNs and Transformers. The results demonstrate the superiority of our method over state-of-the-art competitors.

## 2 BACKGROUNDS

### 2.1 FAIRNESS IN COMPRESSED MODELS

Model compression has garnered great attention over the past decade, where numbers of methods are developed and successfully deployed, creating substantial values in various applications. However, recent works reveal that behind the good performance on the original tasks, potential bias can be introduced or may be present (Hooker et al., 2019; 2020; Stoychev & Gunes, 2022; Iofinova et al., 2023). These works primarily focus on image classification as a case study, concluding that although the overall accuracy of compressed models is comparable to that of their uncompressed models, the influence of compression on the accuracy of individual classes or specific data groups can be significantly different, potentially leading to unpredictable fairness risks.

Meanwhile, it is worth noting that accuracy usually conflicts with prevailing fairness metrics (Zliobaite, 2015). For instance, in an extreme case, a random classification model may be considered to be absolutely fair, as it treats all classes or groups equally by assigning predictions at random. It is expected that if compressed models bear degradation of accuracy, the fairness of them can sometimes benefit from the trade-off relationship and surpass the uncompressed models. Such phenomenon can be observed in our experiments and previous studies (Zhang et al., 2023). On the other hand, research on language models (Xu & Hu, 2022; Zayed et al., 2024) has shown that model compression can potentially serve as regularization tools, helping to improve fairness with sacrifice of quality in generated contents. These findings highlight the importance of fairness-aware model compression, and compressed models should be evaluated from a Pareto-optimal perspective that jointly considers accuracy and fairness, as advocated by Paganini (2020).

### 2.2 FAIRNESS-AWARE MODEL COMPRESSION

Growing attention has been given in recent years to achieve fairness-aware model compression. Substantial efforts are on enhancing conventional model compression techniques, including knowledge distillation (Blakeney et al., 2021), quantization (Liu et al., 2025), and neural network pruning (Lin et al., 2022; Tang et al., 2023; Zhang et al., 2023; Zayed et al., 2024). Among these, neural network pruning has attracted particular interest due to its clear motivation and competitive performance. However, existing works are primarily on unstructured pruning. For example, Lin et al. (2022) designed a tailored pruning metric based on the importance of weights across different data groups and applied pruning accordingly using a greedy strategy. Tang et al. (2023) leveraged the concept of the well-known Lottery Ticket Hypothesis (Frankle & Carbin, 2019) and identified sparse network initializations that inherently exhibit fairness. Zhang et al. (2023) attached an element-wise mask to the weight matrix of a neural network and induced sparsity in the mask by minimizing an adversarial loss. On the other hand, fairness-aware structured pruning has rarely been explored. Moreover, existing unstructured pruning methods are difficult to be adapted to structured pruning, due to the discrete solution space defined by structural components.

### 2.3 STRUCTURED NEURAL NETWORK PRUNING

Structured neural network pruning targets the removal of regular structural components in neural networks, such as filters in CNNs and attention heads in Transformers (Cheng et al., 2024). Compared with unstructured pruning, structured pruning is considered more practical for accelerating inference on modern hardware and software platforms. Over the past decade, a variety of structured

pruning methods have been proposed (Cheng et al., 2024). Among them, the most widely adopted are criteria-based methods, which rely on predefined heuristics, such as weight magnitude (Li et al., 2017), to assess the importance of structural components. Recently, evolutionary algorithms have also demonstrated great potential to search for solutions to structured pruning (Zhou et al., 2021; Shang et al., 2022). Depending on the architecture of the target model, structured pruning may lead to performance degradation. Consequently, it is often accompanied by fine-tuning and sometimes performed iteratively, in order to improve the performance of the pruned model in practice.

On the other hand, achieving fairness-aware structured pruning is more challenging than unstructured pruning, as the discrete nature of structural components makes it impractical to apply gradient-based techniques such as adversarial training (Zhang et al., 2023). To date, the only prior work on fairness-aware structured pruning (Zayed et al., 2024) targets at pruning attention heads in language models using a greedy strategy. Specifically, it exhaustively evaluates the impact of each head on both perplexity and toxicity by comparing the outcomes of retaining versus removing it, and then directly retains the top-performing heads based on this evaluation. In this work, we propose a general approach for fairness-aware structured pruning based on evolutionary optimization. To address the inherent challenges, we incorporate cooperative coevolution (Ma et al., 2019) and multi-objective evolution (Zhou et al., 2011) techniques that have also been employed for normal structured pruning (Shang et al., 2022; Zhou et al., 2021).

## 3 FAIRNESS-AWARE PRUNING VIA EVOLUTION (F-PvE)

In this work, we propose a general fairness-aware structured neural network pruning method based on evolutionary optimization, named F-PvE. From the view of structured pruning, a neural network consists of a number of structured components (e.g., filters in CNNs and attention heads in Transformers), which serve as the basic unit for pruning. Considering this, structured pruning can naturally be formulated as a subset selection problem, which can be solved with evolutionary optimization techniques (Shang et al., 2022). Let $\Phi_C$ denote the neural network to be pruned, where $C$ denotes the set of structured components in $\Phi_C$. Structured pruning aims to identify the optimal subset $C^* \subset C$, such that

$$C^* = \arg\max_{C' \subset C} \mathcal{M}(C'),$$

where $\mathcal{M}(\cdot)$ denotes the performance evaluation of $C'$. Typically, $\mathcal{M}(C')$ is designed to assess the performance of a neural network constructed from $C'$ on specific metrics, i.e., accuracy or fairness calculated on prepared dataset. With explicit evaluation $\mathcal{M}(\cdot)$, evolutionary algorithms can be easily adapted to solve the subset selection problem, which have shown impressive performance from both theoretical and practical perspectives (Qian et al., 2015). Specifically, it maintains a population of individuals where each individual represents a subset $C' \subset C$ (i.e., a pruning solution). We represent each subset $C' \subset C$ with a binary vector made of 0 and 1 with length equal to $|C|$, where each bit indicates whether a specific unit in $C$ is retained or pruned.

A standard evolutionary process proceeds as follows: the population is updated iteratively, through the reproduction of new individuals via mutation, and environmental selection. For mutation, a parent individual is randomly selected from the population. Each bit in the parent vector is flipped with a predefined probability (i.e., the mutation rate), generating a new child individual. To ensure a controllable pruning ratio, the resulting binary vector is adjusted so that the proportion of zeros matches the target. Specifically, if the proportion of zeros exceeds the target, a number of zero bits are randomly flipped to ones; conversely, if it falls below the target, one-bits are flipped to zeros. Typically, in each generation, a number of child individuals equal to the population size are produced. For environmental selection, all parent and child individuals are ranked based on the evaluation $\mathcal{M}(\cdot)$. The top-performing individuals are retained, while the rest are discarded. After a number of generations, the final population serves as the solutions.

Note that the evaluation $\mathcal{M}(\cdot)$ plays a crucial role, which guides the direction of evolution. We can achieve fairness-aware structured pruning by incorporating fairness with accuracy in $\mathcal{M}(\cdot)$ simultaneously. A straightforward but effective approach is to use a weighted sum of accuracy and fairness, which is common for multi-objective problems and shows good practical performance. However, using a predefined weight means that the evolution process is towards a specific point on the trade-off spectrum, which usually requires trial-and-error in practice to meet user preferences. To address this issue, we adopt the well-known multi-objective evolutionary algorithm NSGA-II (Deb et al.,

2002), aiming to obtain a set of Pareto-optimal solutions with respect to accuracy and fairness. This allows users to select from Pareto-optimal solutions based on their preference after evolving. Meanwhile, the large number of candidate basic units for pruning poses challenges for efficient evolution. To overcome this, we employ cooperative coevolution (Ma et al., 2019), which applies a divide-and-conquer strategy to partition the search space, and evolves each subspace in an cooperative manner, enabling more effective exploration. In the following subsections, we will first present the fairness evaluation under three typical fairness scenarios considered in this work. Subsequently, we will introduce F-PvE with multi-objective evolution and cooperative coevolution.

### 3.1 Assessment of Fairness

#### 3.1.1 Class-wise degradation in multi-class classification

In multi-class classification, the models are employed to predict the correct class from multiple alternatives. Hooker et al. (2019) indicated that model compression can result in disparate impact on different classes. That is, the accuracy on some of the classes can degrade disastrously, while the accuracy on some of the classes can maintain or even be improved after pruning. To quantify the disparate impact, we define Extremum of disparate Impact (EDI), the extremum difference of accuracy changes among classes. Denote the set of classes as $S$, and the accuracy on a specific class $s \in S$ of the original unpruned model and pruned model as $\mathcal{A}_o(s)$ and $\mathcal{A}_p(s)$ respectively, we calculate EDI as

$$\text{EDI} = \max_{s \in S}\left(\frac{\mathcal{A}_p(s) - \mathcal{A}_o(s)}{\mathcal{A}_o(s)}\right) - \min_{s \in S}\left(\frac{\mathcal{A}_p(s) - \mathcal{A}_o(s)}{\mathcal{A}_o(s)}\right).$$

#### 3.1.2 Group-wise degradation in classification with sensitive groups

Taking hiring decisions as an example, machine learning models are often employed to predict whether a candidate should be accepted based on their features, which constitutes a typical binary classification task. However, as sensitive attributes such as gender or race may be present in the data, the model may inadvertently favor certain groups, leading to fairness concerns. In other words, we aim to ensure that the model's decisions are independent of these sensitive attributes. To quantify the discrepancy among groups, we employ the prevailing metric Difference of Equalized Odds (DEO) (Hardt et al., 2016) to measure the fairness degree of models. Consider binary classification with two sensitive groups. Denote the sensitive class as $A \in \{0, 1\}$, the target class as $Y \in \{0, 1\}$, and the predicted results as $\widehat{Y} \in \{0, 1\}$, we can calculate DEO as

$$\text{DEO} = \sum_{y \in \{0,1\}} \frac{1}{2}\left|\Pr(\hat{Y} = 1 | A = 0, Y = y) - \Pr(\hat{Y} = 1 | A = 1, Y = y)\right|.$$

#### 3.1.3 Social bias in generated contents of language models

Language models have achieved remarkable success in recent years. However, the generated content may involve underlying social bias, which can lead to severe fairness issues (Dixon et al., 2018). To quantify the bias in language models, we employ the definition of Toxicity (Dhamala et al., 2021). That is, the content is considered to be with toxicity if it leads individuals to disengage from a discussion. Following the implementations in prior works that studied fairness in pruned language models (Zayed et al., 2024), we employ BERT for toxicity assessment, and measure the bias of language models by calculating the Discrepancy of Toxicity (DT) in groups. Denote the groups as $G$, and the toxicity of a specific group $g \in G$ as $tox(g)$, We can calculate DT as

$$\text{DT} = \sum_{g \in G} \frac{1}{|G|}\left|tox(g) - \frac{1}{|G|}\sum_{g' \in G} tox(g')\right|.$$

### 3.2 F-PvE with Multi-objective Evolution

To address problems with multiple objectives, a common strategy is using a weighted sum of the objectives, which is simple yet effective. However, determining an appropriate weight value often requires tedious trial-and-error tuning according to user preferences. To overcome this limitation,

we adopt multi-objective evolution. Specifically, we employ the widely used NSGA-II (Deb et al., 2002), which has demonstrated strong performance in tasks with two objectives. The main mechanism is that, we use non-dominated sorting and crowding distance in environmental selection. Given the individuals for selection, all non-dominated ones are moved to the population of next generation, and this process continues until the population size is reached. If it exceeds the population size, they are further ranked based on their crowding distance, computed as the sum of distances to their neighboring individuals in the objective space. In addition, we employ binary tournament selection to choose parents for mutation, which randomly samples two individuals, selects the better one, aiming to improve efficiency of evolution. The procedure is outlined in Algorithm 1.

---

**Algorithm 1** F-PvE with multi-objective evolution

---

**Input**: The original model $\Phi_C$ to be pruned
**Parameter**: Population size $N$; number $G$ of generations; mutation rate $r$
**Output**: A set $S$ of Pareto-optimal pruned models

1: Initialize the population by randomly sampling $N$ individuals, and evaluate their accuracy and fairness;
2: **while** $G$ is not reached **do**
3:     Generate $N$ child individuals via binary tournament parent selection and mutation, and evaluate them;
4:     Leverage non-dominated sorting and crowding distance to select $N$ individuals from the parent and child individuals, as the next generation of population
5: **end while**
6: **return** the Pareto-optimal set $S$ of the final population

---

### 3.3 F-PvE with Cooperative Coevolution

Since the number of structured units in deep neural networks can be exceedingly large (e.g, in ResNet50 (He et al., 2016), there exist over ten thousand candidate filters to be pruned), how to effectively evolve poses great challenges. To address it, we introduce cooperative coevolution (Ma et al., 2019) in F-PvE, which has been applied in normal structured pruning and shown good performance (Shang et al., 2022). Cooperative coevolution leverages a divide-and-conquer strategy that organizes the structural units into distinct groups, with each group evolving in a cooperative manner. That is, each group focuses on the decision of retaining and pruning on its assigned units, which represents a part of the neural network. For the evaluation of a given individual within a group, the pruned partial network corresponding to that individual is combined with the unpruned partial networks associated with other groups, resulting in a complete neural network that can be used to assess performance. This strategy facilitates efficient local search within each group. Note that a proper grouping is crucial, as units with competitive relationships should be organized into the same group. To address this, we employ DepGraph (Fang et al., 2023), a general approach to identify the relationship among structural units, and apply grouping accordingly. The procedure of F-PvE with cooperative coevolution is outlined in Algorithm 2. Specifically, for the convenience of combining solutions in each group, we simply define the fitness of an individual as a weighted sum of accuracy and fairness, such that we can easily identify the best solutions in each group for combination. Each group evolves via a standard evolution process, through iterative reproduction of new individuals via mutation, combined with an elite environmental selection strategy that greedily retains the top-performing individuals.

---

**Algorithm 2** F-PvE with cooperative coevolution

---

**Input**: The original model $\Phi_C$ to be pruned
**Parameter**: Number $G$ of generations for evolving subgroups; population size $N$; mutation rate $r$
**Output**: The pruned model $\Phi_{C^*}$

1: Divide the structural units of $\Phi_C$ into groups;
2: Parallelly apply evolution process for $G$ generations in each group, with population size $N$ and mutation rate $r$;
3: Select the individual with the best fitness value in each group, and integrate them into a pruned model $\Phi_{C^*}$
4: **return** the pruned model $\Phi_{C^*}$

---

## 4 EXPERIMENTS

To examine the effectiveness of the proposed F-PvE, we conduct experiments on three typical fairness problems, including class-wise degradation in multi-class classification, group-wise degradation in classification with sensitive groups, and social bias in generated contents of language models. Given the distinct nature of these tasks, we organize the section by task. Due to space limitations, we provide our additional results in the Appendix, and summarize the contents and conclusions at the end of this section. Our codes for reproducing the results are provided in the supplementary.

### 4.1 TASK 1: CLASS-WISE DEGRADATION IN MULTI-CLASS CLASSIFICATION

**Settings.** We evaluate F-PvE on two common settings for structured pruning: ResNet-56 on CIFAR-100 (Krizhevsky, 2009) and ResNet-50 on CUB-200 (Wah et al., 2011), with convolution filters as basic units. Due to the large search spaces (2,032 and 22,720 filters, respectively), we apply F-PvE with cooperative coevolution. The top-1 classification accuracy of the model and fairness measured by EDI serve as the objectives during pruning, and we assign a weight $\lambda$ to balance accuracy and fairness. That is, the fitness score of a pruned model is defined as: $(\mathcal{A}_p - \mathcal{A}_o)/\mathcal{A}_o - \lambda \cdot \mathrm{EDI}$, where $\mathcal{A}_o$ and $\mathcal{A}_p$ denote the top-1 accuracy of the original and pruned model, respectively, and EDI is subtracted as it should be minimized. Both accuracy and EDI are evaluated on the training set of the corresponding task during evolution. The population size $N$, number $G$ of generations and mutation rate $r$ are set as 5, 10 and 0.05, respectively, which can be increased to potentially achieve better performance with additional search budgets. For comparison, we follow previous work on fairness-aware structured pruning and implement prevailing structured pruning methods based on criteria, including the weight magnitude $l_1$ and $l_2$ norm (Li et al., 2017), Hessian (LeCun et al., 1989), and FPGM (He et al., 2019). All pruned models are finetuned on the training set under the same settings (details are provided in Appendix). On CIFAR-100, we use iterative pruning, removing 5% of the convolution filters and finetuning the pruned network in each iteration, while on CUB-200, we adopt one-shot pruning without iteration.

Table 1: Comparison results on Task 1, where 20% of the structural components are pruned (**Bold** indicates the best).

(a) ResNet56 on CIFAR-100 (Iterative pruning)

| Method | ACC%↑ | EDI↓ |
|---|---|---|
| *Original Model* | 72.78 | – |
| Random | 69.30 | 0.4136 |
| $l_1$ | 69.55 | 0.3942 |
| $l_2$ | 69.72 | 0.4178 |
| Hessian | 69.49 | 0.4105 |
| FPGM | 69.71 | 0.4548 |
| F-PvE ($\lambda$=0) | **69.86** | 0.4636 |
| F-PvE ($\lambda$=0.01) | 69.79 | **0.3938** |

(b) ResNet50 on CUB-200 (One-shot pruning)

| Method | ACC%↑ | EDI↓ |
|---|---|---|
| *Original Model* | 84.81 | – |
| Random | 83.09 | 0.6609 |
| $l_1$ | 83.33 | 0.6855 |
| $l_2$ | 83.18 | 0.5839 |
| Hessian | 83.30 | 0.6224 |
| FPGM | 83.25 | 0.6937 |
| F-PvE ($\lambda$=0) | **83.58** | 0.6301 |
| F-PvE ($\lambda$=0.01) | 83.39 | **0.5743** |

**Analysis.** The main results of pruned models on test sets are provided in Table 1, where the values are average of four independent runs since evolution and finetuning can both involve randomness. On CIFAR-100, F-PvE with $\lambda = 0$ (i.e., optimizing accuracy only) achieves the highest top-1 accuracy of 69.86%, but also results in the worst EDI of 0.4636, as fairness is not considered during evolution. In contrast, F-PvE with $\lambda = 0.01$ achieves a slightly lower accuracy of 69.79%, but significantly improves fairness with the best EDI of 0.3938. Mean-

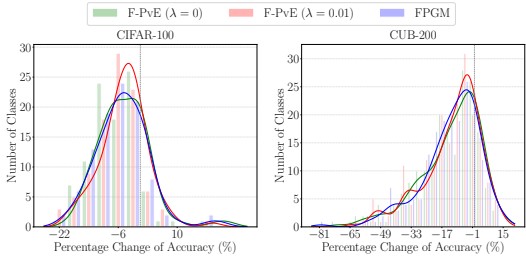

Figure 1: Class-wise accuracy change on Task 1.

while, compared to the baselines, F-PvE with $\lambda = 0.01$ achieves better accuracy and fairness simultaneously, demonstrating its superiority. Similar results are observed on CUB-200. To further analyze class-wise accuracy changes, we visualize the number of classes across different levels of percentage change in accuracy in Figure 1. As expected, F-PvE with $\lambda = 0.01$ leads to smaller discrepancies among classes, resulting in better fairness.

## 4.2 TASK 2: GROUP-WISE DEGRADATION IN CLASSIFICATION WITH SENSITIVE GROUPS

**Settings.** We examine F-PvE by pruning ResNet-18 on CelebA (Liu et al., 2015), a widely used benchmark to study fairness-aware model compression (Zhang et al., 2023). To emphasize the impact of pruning, we halve the architecture and randomly shrink the dataset to $10\%$ (details are in the Appendix). *Attractive* is set as the target class and *Gender* is set as the sensitive class. We leverage the top-1 accuracy on target class and fairness metric DEO for evaluation. As it also involves a large-scale search space (i.e, 3904 filters for ResNet18), we employ F-PvE with cooperative coevolution. Similar to Task 1, the fitness of a pruned neural network during evolution is computed as a weighted sum $(\mathcal{A}_p - \mathcal{A}_o)/\mathcal{A}_o - \lambda \cdot (\mathrm{DEO}_p - \mathrm{DEO}_o)/\mathrm{DEO}_o$, which is evaluated on the training set of the task. The same to Task 1, the population size $N$, number $G$ of generations and mutation rate $r$ are set as 5, 10 and 0.05, respectively. For comparison, we implement $\ell_1$, $\ell_2$, Hessian, and FPGM. All methods adopt iterative pruning, removing $10\%$ convolution filters of each layer and finetuning the pruned network in each iteration, until $40\%$ of filters are pruned.

Table 2: Results of pruning ResNet-18-Half on CelebA in Task 2 (**bold** indicates the best; underline indicates the runner-up).

| Method | PR = 10% | | PR = 20% | | PR = 30% | | PR = 40% | |
|---|---|---|---|---|---|---|---|---|
| | ACC%↑ | DEO↓ | ACC%↑ | DEO↓ | ACC%↑ | DEO↓ | ACC%↑ | DEO↓ |
| *Original model* | 80.06 | 0.5049 | 80.06 | 0.5049 | 80.06 | 0.5049 | 80.06 | 0.5049 |
| $l_1$ | 79.61 | 0.4753 | 79.42 | 0.4632 | 79.17 | 0.4555 | 79.08 | 0.4637 |
| $l_2$ | 79.64 | **0.4665** | 79.43 | 0.4608 | 79.21 | 0.4598 | 79.21 | 0.4640 |
| Hessian | 79.62 | 0.4684 | 79.38 | 0.4683 | 79.27 | 0.4653 | 78.96 | 0.4619 |
| FPGM | 79.59 | 0.4723 | 79.42 | 0.4668 | 79.11 | 0.4572 | 79.16 | 0.4583 |
| F-PvE ($\lambda = 0$) | 79.77 | 0.4798 | 79.46 | 0.4657 | **79.33** | 0.4616 | **79.31** | 0.4633 |
| F-PvE ($\lambda = 0.1$) | **79.81** | 0.4682 | **79.57** | **0.4589** | 79.27 | **0.4532** | 79.29 | **0.4551** |

**Analysis.** The results of pruned models on test sets are presented in Table 2, where each value represents the average of eight independent runs. We can observe that F-PvE with $\lambda = 0.1$ consistently achieves better DEO compared to F-PvE with $\lambda = 0$, demonstrating the effectiveness of incorporating fairness into the fitness evaluation of individuals during evolution. Compared with the competing methods, except for the small pruning ratio (PR) of $10\%$, F-PvE with $\lambda = 0.1$ consistently achieves better accuracy and lower DEO simultaneously.

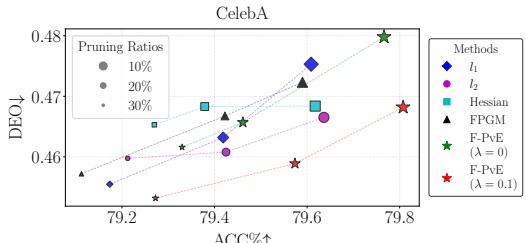

Figure 2: Illustration of comparison on Task 2.

We further present the results in Figure 2 for a clearer comparison. All methods exhibit a consistent trend: higher pruning ratios lead to lower accuracy but improved DEO. This is reasonable due to the inherent trade-off of accuracy and fairness. Meanwhile, we can find that F-PvE with $\lambda = 0.1$ achieves the best balance, with its markers positioned closer to the bottom-right corner.

## 4.3 TASK 3: SOCIAL BIAS IN GENERATED CONTENTS OF LANGUAGE MODELS

**Settings.** Following the only prior work on fairness-aware structured pruning, FASP (Zayed et al., 2024), we evaluate our method under the same settings, which target at social bias in pruned language models. Specifically, we prune attention heads in Transformers. To assess model performance, we use WikiText-2 and report perplexity (PPL) to measure accuracy. For fairness, we adopt DT as the metric. We adopt the open-source implementation of FASP and follow its data split strategy, using a validation set for evaluation of impact on heads or pruned models, and a test set for final performance comparison, which ensures a fair comparison. For evaluation, we prune GPT-2 and DistilGPT-2 (Radford et al., 2019), which contain 144 and 72 candidate attention heads, respectively. Given the relatively small number of structured components, we apply F-PvE with multi-objective evolution. Additionally, we evaluate a variant of F-PvE using a weighted sum of accuracy and fairness as the fitness function, and the results still remain competitive (provided in

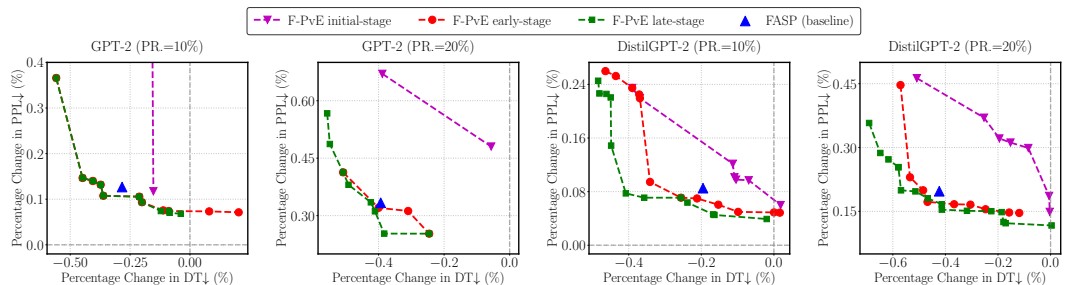

Figure 3: Comparison results on Task 3. The Pareto-optimal solutions are illustrated.

the Appendix), demonstrating the effectiveness of evolution for fairness-aware structured pruning. In the evolution process, the population size $N$ is set to 32, the mutation rate $r$ to 0.1, and the total number $G$ of generations to 25. We present the performance of F-PvE at various stages throughout the evolution for clear comparison.

**Analysis.** As FASP has been shown to outperform criteria-based structured pruning methods (Zayed et al., 2024), we focus the comparison on F-PvE and FASP for clarity. In FASP, the evaluation cost is $3 \times$ HeadsNum, as each head is evaluated three times to reduce the effect of randomness and ensure reliable impact estimation for greedy selection. In contrast, F-PvE evaluates each individual only once, with a cost of $N \times g$, where $g \leq G$ denotes the number of completed generations. Figure 3 illustrated the results. For F-PvE, we illustrate the Pareto-optimal solutions at initial stage (initial generation), middle stage with comparable cost to FASP ($g = 15$ for GPT-2 and $g = 7$ for DistilGPT-2), and final stage ($g = 25$) respectively. We can observe that F-PvE converges fast, showing significant progress from the initial to early stages. At early stage, F-PvE can outperform FASP, yielding solutions superior in both accuracy and fairness. By the final stage, F-PvE exhibits a clear advantage, not only providing a variety of Pareto-optimal solutions, but also obtaining significantly superior solutions (e.g., on DistilGPT-2, with competitive PPL and significantly better DT).

### 4.4 ADDITIONAL RESULTS

Due to space limitations, we place some additional but interesting results in Appendix, summarized as follows: (1) **Additional fairness metrics**: We use the entropy of class-wise accuracy change after pruning as fairness metric, which also demonstrates the superiority of F-PvE. (2) **Effectiveness of weighted sum strategy**: We delve into the weighted sum strategy. Results show that on specific preference controlled by $\lambda$, it can surpass multi-objective evolution; preference can be effectively controlled by $\lambda$. (3) **Combination with fairness-enhancing techniques**: We primarily verify adversarial training for finetuning process on Task 2, showing that the superiority of F-PvE is adaptable to other techniques for improving fairness.

## 5 LIMITATIONS AND DISCUSSIONS

**Limitations.** (1) *Efficiency*. As a search-based approach, F-PvE requires a higher computation budget for pruning compared with classic criteria-based methods, but the overhead remains acceptable as presented in Appendix. Meanwhile, inference budget is more critical in practice, and F-PvE consistently demonstrates superior performance across various pruning ratios. (2) *More scenarios*. The real-world involves various fairness scenarios. With limitations in open-source benchmarks and computation resources, we include three typical fairness scenarios, covering the pruning of CNNs and Transformers. We will explore the applicability to broader scenarios in the future.

**Discussions.** Structured pruning has received considerable attention, yet how to achieve fairness-aware structured pruning remains underexplored. This work proposes the first general fairness-aware structured neural network pruning method via evolution, which we hope can spark further research on this important topic. Moreover, integrating existing fairness techniques (Pessach & Shmueli, 2022) with structured pruning also holds great potential in practice. As our primary goal is highlighting the effectiveness of evolutionary optimization on this task, we only adopt simple operators and frugal hyper-parameter settings. We believe that incorporating advanced evolutionary techniques could further enhance performance, which we leave for future work.

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

## A  OVERVIEW OF THE APPENDIX

The Appendix includes:

- A supplementary introduction to the setup of experiments, including the models for pruning, finetuning, and other details to reproduce the results.
- Results on additional fairness metrics, the effectiveness of weighted sum strategies, and the combination with fairness-enhancing techniques (i.e., adversarial training), as summarized in the paper. Meanwhile, we provide additional results of Task 3, including results on validation set and results of repeated runs.

## B  DETAILED EXPERIMENTAL SETTINGS

The experiments on Task 1 and Task 2 are implemented from scratch in Pytorch. For Task 3, we implement our method and baseline method based on the official implementation of the prior work FASP (https://github.com/chandar-lab/FASP). The implementation of DepGraph in F-PvE with cooperative coevolution is adopted from its official repository (https://github.com/VainF/Torch-Pruning). The implementation of the comparison methods $l_1$, $l_2$, Hessian and FPGM is based on https://github.com/VainF/Torch-Pruning as well. All of our experiments are conducted on an Ubuntu server equipped with two NVIDIA RTX A6000 Ada Generation GPUs. The detailed introduction on each task is provided below.

### B.1  TASK 1: CLASS-WISE DEGRADATION IN MULTI-CLASS CLASSIFICATION

**ResNet-56 on CIFAR-100**

*The model for pruning*: We train the model from scratch for 200 epochs using SGD as the optimizer, with initial learning rate as 0.1, weight decay as 0.0001, momentum as 0.9, and batch size as 128. The learning rate is decayed at epochs 120, 150, and 180 using a multi-step schedule. We further apply an early-stop strategy to select the best model for pruning. The data augmentation pipeline includes random cropping (32×32 pixels with 4-pixel padding), horizontal flipping (50% probability), and normalization (with mean as [0.5071, 0.4867, 0.4408] and standard deviation as [0.2675, 0.2565, 0.2761]).

*Finetuning*: During finetuning, we use the same data augmentation strategy as that for training the model for pruning, but adjust the learning rate to 0.01, increase the batch size to 256, and finetune for 100 epochs. For a fair comparison, in the implementation of F-PvE and all the competitors, we do not apply early-stop strategy.

*Other details*: We prune each layer in the neural networks equally, with the the same proportion equal to the pruning ratio. For iterative pruning, in each iteration, we prune the neural network and finetune the resulting model. The finetuned model then serves as the neural network for pruning in the next iteration. We conduct four independent runs in the experiments, with seeds as {2021, 2022, 2023, 2024}. Each single run can be finished within 12 hours on a single GPU (note that the time is not strictly precise as it depends on the server load).

**ResNet-50 on CUB-200**

*The model for pruning*: We leverage the pretrained model on ImageNet as the starting point, and further train it for 88 epochs, with initial learning rate as 0.002, weight decay as 0.0001, momentum as 0.9, and batch size as 8. We employ a cosine annealing learning rate scheduler. We further apply the early-stop strategy to select the best model as the model for pruning. For data augmentation, we resize images to 512×512 pixels, followed by random cropping to 448×448 pixels and horizontal flipping (with 50% probability), with normalization using ImageNet-standard parameters (with mean as [0.485, 0.456, 0.406] and standard deviation [0.229, 0.224, 0.225]).

*Finetuning*: During finetuning, we use the same data augmentation strategy and training settings as those used in training the model for pruning. The finetuning is conducted for 50 epochs. For a fair comparison, in the implementation of F-PvE and all competing methods, we do not apply early-stop strategy.

*Other details*: We prune each layer in the neural networks equally. Note that we do not use iterative pruning on this task, since it can lead to severe degradation of accuracy. A possible reason is that the size of CUB-200 is relatively small, and we use pretrained weights on ImageNet in the model for pruning to improve accuracy, which is, however, not well-suited to the iterative pruning manner. We applied four single runs in the experiments, with seeds as {2021, 2022, 2023, 2024}. Each single run can be finished within 10 hours on one GPU as provided.

### B.2    TASK 2: GROUP-WISE DEGRADATION IN CLASSIFICATION WITH SENSITIVE GROUPS

In task 2, we prune ResNet-18 on CelebA, a widely used benchmark to study the group-wise fairness in classification with sensitive groups. Nevertheless, we observed that directly applying structured pruning on original ResNet-18 on CelebA results in only minimal degradation of performance, suggesting that the dataset and neural network themselves are much too redundant. To verify the effectiveness of different structured pruning methods, we increase the difficulty of the task with a simple modification. Specifically, we reduce the training data size to 10% through random sampling, and slim ResNet-18 by halving the number of convolution filters in each layer. Meanwhile, we choose *Attractive* as the target class and *Gender* as the sensitive class, which is widely used in previous studies, due to its desirable property of class balance. These settings allow for a more accurate assessment of the pruning methods' effectiveness.

*The model for pruning*: The model is trained from scratch for 50 epochs using SGD, with initial learning rate as 0.1, weight decay as 0.0001, momentum as 0.9 and batch size as 256. We apply cosine annealing scheduler to the learning rate. For data augmentation, images are resized to $224 \times 224$ pixels with augmentations including vertical and horizontal flipping (Each one has 50% probability), random rotation ($\pm 15°$), and normalization (with mean as [0.506, 0.426, 0.383] and standard deviation as [0.266, 0.245, 0.241]).

*Finetuning*: We use the same training settings and data augmentation procedure as those used in training the model for pruning, to finetune the pruned models for 20 epochs. Similarly to Task 1, we do not apply early-stop strategy for all the methods in order to ensure a fair comparison.

*Other details*: Similar to Task 1, we prune each layer in the neural networks equally. For iterative pruning, we prune the neural network and finetune the resulting model in each iteration, and the finetuned model then serves as the neural network for pruning in the next iteration. We conduct eight independent runs, with seeds {1, 2, 3, 4, 5, 6, 7, 8}. Each run can be finished within 18 hours on a single GPU (note that the time is approximate as it depends on the server load).

### B.3    TASK 3: SOCIAL BIAS IN GENERATED CONTENTS OF LANGUAGE MODELS

The implementation of Task 3 is largely based on the official repository of the prior work FASP. Specifically, the models used for pruning are obtained from Hugging Face, and the pruned models are directly evaluated on the corresponding test set without finetuning. The required time to produce the results of F-PvE with multi-objective evolution is approximately 2 days and 7 days for DistilGPT-2 and GPT-2 on a single GPU, respectively. Due to limitation of computation resources, we report the results using random seed 1 as the main results in the paper, and we provide results with multiple independent runs on GPT-2, demonstrating that F-PvE consistently outperforms FASP across different seeds.

## C    ADDITIONAL RESULTS

### C.1    ADDITIONAL FAIRNESS METRIC

To verify that our method F-PvE can generalize to broad scenarios and metrics, we conduct experiments on another fairness metric under Task 1, using the one-shot pruning setting that prunes ResNet50 on CUB-200. That is, instead of EDI, we calculate the entropy of class-wise accuracy changes after pruning, and take it as the fairness metric. A lower entropy indicates a more uniform impact across classes, and thus, it should be minimized. At the pruning ratio of 40%, we run F-PvE with entropy and EDI as the fairness metric, respectively. The results are provided in Table 3 and 4, which demonstrate that compared to the competing methods, F-PvE achieves the best fairness

with competitive accuracy under both fairness metric. Moreover, we can observe that the rankings among criteria-based methods on different settings can vary significantly. For example, FPGM ranks second under EDI, but performs best on entropy; Hessian achieves the best performance under EDI but ranks lowest under entropy. These findings further underscore the importance of explicitly incorporating fairness during pruning.

Table 3: Comparison results (Entropy) of ResNet-50 on Cub-200 with PR = 40%. **Bold** indicates the best.

| Method | ACC%↑ | Entropy↓ |
|---|---|---|
| *Original Model* | 84.81 | - |
| $l_1$ | 82.33 | 1.7705 |
| $l_2$ | 82.42 | 1.8061 |
| Hessian | 82.22 | 1.8056 |
| FPGM | 82.41 | 1.7350 |
| F-PvE ($\lambda = 0$) | **82.85** | 1.8009 |
| F-PvE ($\lambda = 0.01$) | 82.79 | **1.7094** |

Table 4: Comparison results (EDI) of ResNet-50 on Cub-200 PR = 40%. **Bold** indicates the best performance.

| Method | ACC%↑ | EDI↓ |
|---|---|---|
| *Original Model* | 84.81 | - |
| $l_1$ | 82.23 | 0.6807 |
| $l_2$ | 82.42 | 0.6804 |
| Hessian | 82.22 | 0.5932 |
| FPGM | 82.41 | 0.6329 |
| F-PvE ($\lambda = 0$) | **82.85** | 0.6697 |
| F-PvE ($\lambda = 0.01$) | 82.78 | **0.5614** |

## C.2 EFFECTIVENESS OF WEIGHTED SUM STRATEGY

For problems involving multi-objectives, the weighted sum strategy is a commonly used approach that can achieve good performance on specific preference controlled by the weight value. However, the tuning of weight value according to user preference can be tedious and non-trivial, which can be addressed by multi-objective optimization techniques that directly return a set of Pareto-optimal solutions. Since F-PvE with cooperative coevolution is not compatible to multi-objective evolution techniques due to the conflict of the divide-and-conquer strategies and the inherent behavior of maintaining Pareto-optimal solution set, we adopt the weighted sum strategy to optimize accuracy and fairness simultaneously in F-PvE with cooperative coevolution. In this subsection we aim to address two research questions. *Q1. Can the weighted sum strategy achieve competitive performance compared to multi-objective evolution for a specific preference? Q2. Can the user preference be effectively controlled by adjusting the weight value $\lambda$?* For *Q1*, we conduct additional experiments on Task 3, implementing F-PvE with the weighted sum strategy and comparing it to F-PvE with multi-objective evolution. For *Q2*, we investigate the effect of different $\lambda$ settings on Task 1.

*Q1. Can the weighted sum strategy achieve competitive performance compared to multi-objective evolution for a specific preference?* We conduct experiments on GPT-2 and DistilGPT-2 with pruning ratios as 10% and 20%, respectively. For the weighted sum strategy, we adopt a standard evolutionary algorithm with population size as 10 and mutation rate as $0.1$. On GPT-2 and DistilGPT-2, it evolves for 20 generations and 10 generations, respectively, where the cost for the evaluation of accuracy and fairness is even less than half of that for the prior work FASP. For comparison, we also illustrate the Pareto-optimal sets obtained by F-PvE with multi-objective evolution (to ensure a comparable evaluation budget with FASP, we use the solutions from generation 15 for GPT-2, and generation 5 for DistilGPT-2). The results on validation set (used for evaluating the impact of attention heads during pruning) and test set are shown in Figure 4 and Figure 5. The red line represents

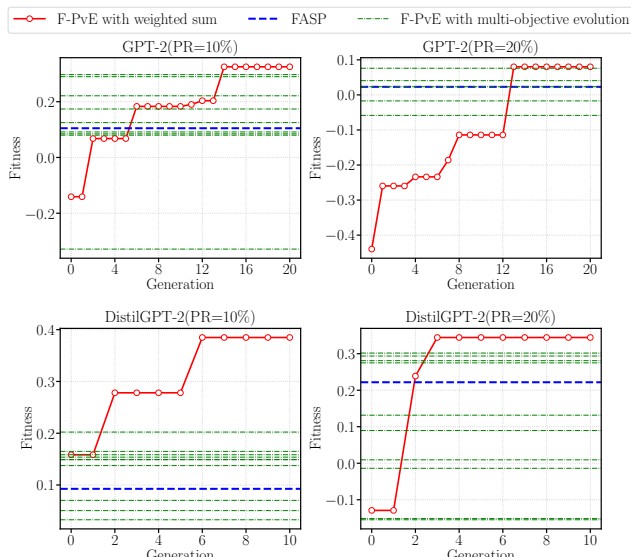

Figure 4: Comparison results of weighted sum strategy and multi-objective evolution on fitness calculated with $\lambda = 0.1$. The illustrated results are on the validation set.

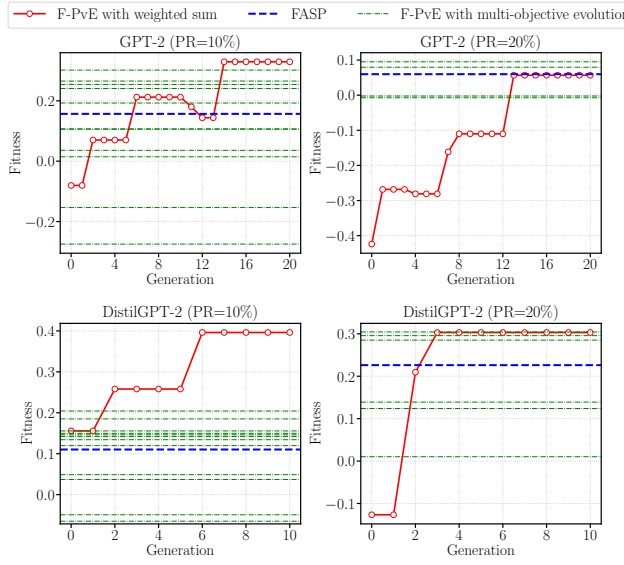

Figure 5: Comparison results of weighted sum strategy and multi-objective evolution on fitness calculated with $\lambda = 0.1$. The illustrated results are on the test set.

the fitness of best individual in the population, for F-PvE with the weighted sum strategy. Each green dash line represents the fitness of a solution in the Pareto-optimal set of F-PvE with multi-objective evolution. The blue dash line represents the fitness of FASP. These results demonstrate that the weighted sum strategy is effective in optimizing both accuracy and fairness within F-PvE, achieving superior pruning results for specific preferences while requiring significantly fewer evaluation resources.

*Q2. Can the user preference be effectively controlled by adjusting the weight value $\lambda$?* To verify this, we run F-PvE with the weighted sum strategy under different values of $\lambda$ (0, 0.01, and 0.1). The results are in Table 5, where we observe that as $\lambda$ increases, accuracy gradually decreases while fairness improves. This shows the trade-off between accuracy and fairness can be effectively controlled by $\lambda$.

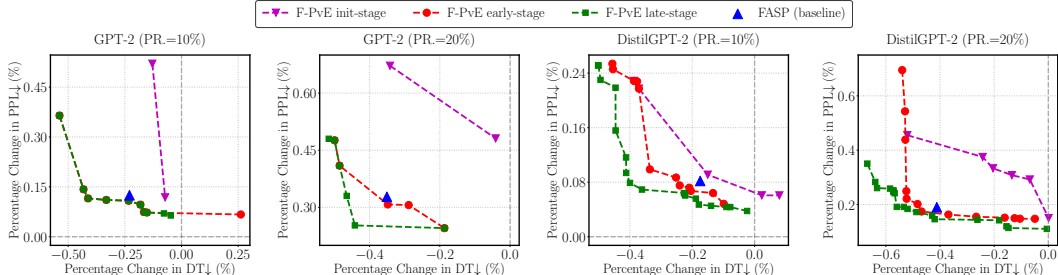

Figure 6: Comparison results (validation set) on Task 3. The Pareto-optimal solutions of F-PvE at different stages are illustrated.

Table 5: $\lambda$ Comparison results on ResNet-56 (CIFAR-100) PR=20%. **Bold** indicates the best performance.

| Method | ACC%↑ | EDI↓ |
|---|---|---|
| *Original Model* | 72.78 | - |
| F-PvE ($\lambda = 0$) | **69.86** | 0.4636 |
| F-PvE ($\lambda = 0.01$) | 69.79 | 0.3938 |
| F-PvE ($\lambda = 0.1$) | 69.74 | **0.3850** |

### C.3 COMBINATION WITH FAIRNESS-ENHANCING TECHNIQUES

There are various methods proposed to enhance the fairness of models from different perspectives, which can potentially be combined to further improve the fairness of pruned models. Here we leverage adversarial training (Madras et al., 2018) as an example. Adversarial training is one of the most popular methods to address fairness issues in classification with sensitive classes (Task 2). The core idea is to learn fair feature representations during training, by attaching an auxiliary classifier to the feature representations and using its gradient information to reduce potential sensitive class information. This process encourages the learned feature representations to be independent of the sensitive classes, thus improving fairness. Details of the implementation are available in our code. Note that a significant drawback of adversarial training is that it is very sensitive to hyperparameter settings. Therefore, our implementation is a preliminary exploration. We employ the Adam optimizer with an initial learning rate of 0.0001, along with a cosine annealing learning rate scheduler. The loss weight of classifier for the target classes and sensitive classes is set as $w = 0.5$. The results are provided in Table 6, which reports the average performance over four independent runs with random seeds {2021, 2022, 2023, 2024}. From the results, we observe that adversarial training significantly improves fairness compared to our main results in the paper, with acceptable sacrifice on accuracy. Meanwhile, F-PvE is still effective, as it enables better trade-off between accuracy and fairness.

Table 6: Debias Comparison results on ResNet-18-Half (CelebA) PR = 20%. **Bold** indicates the best performance.

| Method | ACC%↑ | DEO↓ |
|---|---|---|
| *Original Model* | 80.06 | 0.5049 |
| $l_1$ | 78.75 | 0.2958 |
| F-PvE ($\lambda = 0$) | **79.21** | 0.2976 |
| F-PvE ($\lambda = 0.1$) | 79.16 | **0.2832** |

## C.4 ADDITIONAL RESULTS ON TASK3

**Results on the validation set**

In the main paper, we report the results of Task 3 on the test set. Here, we additionally provide the corresponding results on the validation set for interested readers. Note that the validation set is used in F-PvE to evaluate individuals during the evolutionary process, and is also utilized in the prior work FASP to assess the impact of individual attention heads. As shown in the results, the performance on the validation set is generally consistent with the test set.

**Results of repeated runs**

Considering that illustrating the Pareto-optimal solutions of multiple stages and multiple runs simultaneously can make the visualization disordered, we only illustrate the results of multiple stages of one single run with seed as 1, in our main paper. Here we illustrate the Pareto-optimal solutions of multiple runs but a single stage (identified by the number of generations) with seed 1, 2, 3. Due to limited computational resources, we use GPT-2 with a pruning ratio of 0.2 as a representative case study. The results on validation set and test set are provided in Figure 7. We can observe that after evolving for 25 generations, all runs surpass the baseline method FASP by obtaining a frontier constructed by the set of Pareto-optimal solutions that can cover FASP, or achieve competitive performance with a frontier that lies close to FASP. Furthermore, when we enlarge the number of generations to 50, we find that the performance of F-PvE can be further improved, demonstrating that additional evaluation budgets can lead to better trade-offs between accuracy and fairness.

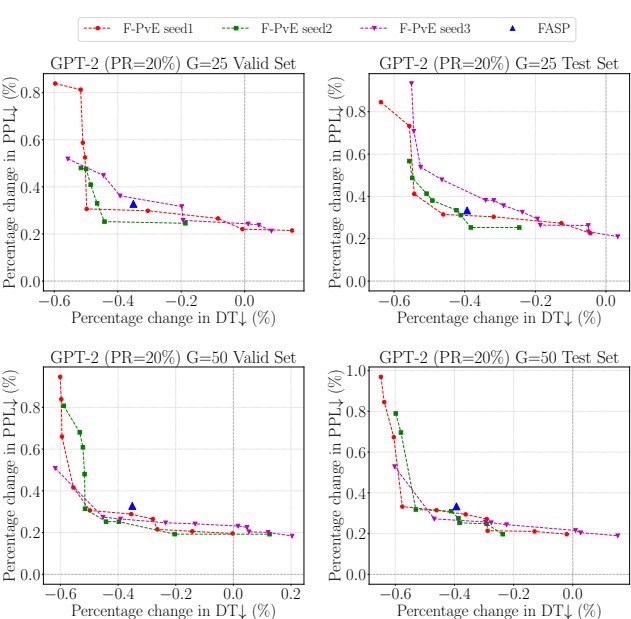

Figure 7: Results of multiple independent runs on Task 3.

