# OpenReview forum: "F-PvE: Fairness-Aware Structured Neural Network Pruning via Evolution"
_ICLR.cc/2026/Conference — ICLR 2026 Conference Withdrawn Submission_

### Official Review · Reviewer_5ihw · 2025-10-27

**Soundness:** 2
**Presentation:** 2
**Contribution:** 2
**Rating:** 4
**Confidence:** 3

**Summary:**

This paper addresses the problem of fairness degradation in model compression, particularly within the context of structured neural network pruning. The authors propose F-PvE, a unified evolutionary optimization framework designed to achieve fairness-aware structured pruning. The method formulates pruning as a subset selection problem and optimizes both accuracy and fairness simultaneously. Two variants are introduced: multi-objective evolution and cooperative coevolution. Experiments are conducted on three types of fairness-related tasks: (1) class-wise fairness on CIFAR-100 and CUB-200, (2) group-wise fairness on CelebA, and (3) social bias mitigation in language models using GPT-2 and DistilGPT-2.

**Strengths:**

1. The paper clearly identifies fairness degradation as a critical issue in model compression and effectively formulates structured pruning as a multi-objective problem involving both accuracy and fairness.
2. The experiments cover multiple architectures and fairness settings, including CNNs and Transformers, demonstrating the generality and adaptability of the proposed framework.

**Weaknesses:**

1. While the paper introduces an evolutionary approach to fairness-aware structured pruning, the core optimization framework relies heavily on well-established methods such as NSGA-II and cooperative coevolution. The work mainly adapts these existing algorithms to the fairness context without providing new algorithmic insights or innovations. As a result, the contribution may appear incremental rather than conceptually novel.

2. The method relies heavily on hyperparameter tuning, particularly the weight λ in the fitness function. Different tasks and models may require different λ values to balance accuracy and fairness, which introduces additional effort and uncertainty in practical applications.

3. The computational cost of F-PvE is significantly higher than that of criteria-based pruning methods. This is because evolutionary algorithms require iterative evaluation of multiple candidate models throughout the search process, which is computationally intensive. Although the paper acknowledges the high computational cost of evolutionary search and proposes cooperative coevolution to improve efficiency, it provides only a qualitative discussion.

**Questions:**

- Could the authors clarify what aspects of F-PvE are fundamentally novel beyond adapting these standard approaches to a fairness-aware pruning setting? It would be helpful to elaborate on whether any new search operators, objective formulations, or convergence criteria are specifically designed for fairness optimization.
- Could the authors elaborate on how λ is selected for different experiments and whether its value significantly affects performance stability?
- While the paper mentions that cooperative coevolution improves scalability, the evaluation mainly remains qualitative. Could the authors provide concrete runtime comparisons with baseline pruning methods or ablation studies showing how cooperative coevolution reduces the search cost?

---

### Official Review · Reviewer_yWHp · 2025-10-28

**Soundness:** 2
**Presentation:** 3
**Contribution:** 2
**Rating:** 4
**Confidence:** 4

**Summary:**

The paper studies fairness-aware structured pruning and proposes F-PvE, a general evolutionary-optimization framework that treats structured pruning as a subset-selection problem over architectural units (e.g., CNN filters, Transformer heads). Each candidate solution encodes which units to keep/prune in a binary vector; evolution (mutation + environmental selection) then optimizes an evaluation functional 𝑀(⋅) that combines accuracy and fairness on a validation set. For models with few components (e.g., heads), F-PvE uses multi-objective evolution (NSGA-II) to balance accuracy/fairness; for models with many components (e.g., CNN filters), it uses cooperative co-evolution to scale search. Experiments span three “fairness scenarios”: class-wise degradation in multi-class classification, group-wise degradation with sensitive attributes, and toxicity/social bias for LMs; the paper reports superior trade-offs versus standard structured pruning and the one prior head-pruning fairness method. The paper positions itself as the first general approach to fairness-aware structured pruning (prior work handled only Transformer head toxicity via a greedy method) and motivates the need by citing fairness harms introduced by compression.

**Strengths:**

1- The paper directly addresses an emerging pain point: compression can skew error across classes/groups or amplify toxicity; structured pruning is especially relevant because it maps well to real hardware speedups.

2- Casting fairness-aware structured pruning as evolution over subsets with an explicit multi-objective is clean and implementable; the binary encoding and controllable pruning-ratio step are practical touches.

3- Using NSGA-II when the unit count is small and co-evolution for large unit counts is a sensible, documented design that aids scalability.

4- The paper claims coverage of class-wise, group-wise, and LM toxicity scenarios across CNNs and Transformers, which (if fleshed out) would be a strong empirical contribution.

**Weaknesses:**

1- The core idea (evolutionary search optimizing a fairness-aware objective) is a predictable extension of existing structured-pruning via evolution; the paper needs stronger contrasts vs. (i) greedy head-pruning for toxicity and (ii) fairness-aware training/regularization and constrained ERM (e.g., Fairlearn/FairReg/FairRet/SA methods) to demonstrate added value beyond “search with a different objective.”

2- It’s unclear which fairness metrics are optimized/measured in each scenario (e.g., class-balance error, worst-group accuracy, demographic parity, equalized odds, toxicity rate). Provide metric definitions, calibration of thresholds, and statistical significance across multiple seeds. Without this, “consistently outperforms” is hard to verify.

3- Evolutionary search can be compute-intensive (population × generations × fine-tuning). The paper should fix a wall-clock or FLOP budget and compare to (a) non-evolutionary structured pruning with fairness-aware fine-tuning, (b) joint training with fairness constraints (e.g., group-loss constraints), and (c) lottery-ticket style selection with fairness-guided saliency at matched cost.

4- Add ablations for (i) NSGA-II vs. weighted-sum scalarization, (ii) with/without co-evolution, (iii) mutation rate, population size, generation count, (iv) influence of fine-tuning schedule per candidate. This will show where the gains come from rather than attributing them to the umbrella “F-PvE.”

5- Evaluate sensitivity to dataset shift and class/group imbalance, and report worst-group performance with confidence intervals. In the LM toxicity setting, consider prompt distribution shifts and disentangle fairness from perplexity degradation to rule out “fairness by uniformly worse language modeling.”

**Questions:**

1- How exactly is 𝑀(⋅) defined per scenario? Do you optimize a vector (accuracy,−fairness_gap) with NSGA-II in “few-unit” settings and a scalarized version with co-evolution? Please provide formulas and normalization details so accuracy/fairness are weighed comparably.

2- Which group-fairness metrics do you optimize/measure (DP, equalized odds, worst-group acc., class-balance error)? For LM toxicity, is the toxicity classifier fixed and calibrated? If so, which model/threshold?

3- At matched compute, how do you compare against (a) constrained ERM with group constraints and (b) fairness regularizers during fine-tuning? (These are natural baselines many practitioners would attempt before evolutionary search.)

4- Please add NSGA-II vs. scalarization, co-evolution on/off, population × generations sweeps, and fine-tuning budget sweeps to isolate which ingredients matter.

5- What is the variance across random seeds (initial population, mutation randomness, data splits)? Show error bars on fairness and accuracy for the reported Pareto fronts.

6- For CNNs with thousands of components, what are the population sizes, generations, and wall-clock? How does F-PvE scale to larger ViT/LLM backbones where pruning units are numerous? Co-evolution partition strategy details would help.

---

### Official Review · Reviewer_CPRZ · 2025-10-30

**Soundness:** 2
**Presentation:** 2
**Contribution:** 2
**Rating:** 2
**Confidence:** 4

**Summary:**

The paper propose F-PvE (Fairness-aware Pruning via Evolution) — a general evolutionary-optimization-based framework that jointly optimizes accuracy and fairness during structured pruning. F-PvE formulates pruning as a subset-selection problem and employs multi-objective evolution (NSGA-II) and cooperative co-evolution to handle both multi-objective optimization and the combinatorial search space. The authors evaluate F-PvE on three fairness scenarios: (1) class-wise degradation (CIFAR-100, CUB-200), (2) group-wise degradation (CelebA), and (3) social bias in language models (GPT-2 / DistilGPT-2). Results show that F-PvE improves fairness metrics (EDI, DEO, DT) with minimal loss in accuracy compared to baselines and a prior fairness-aware method (FASP).

**Strengths:**

1. F-PvE creatively formulates fairness-aware structured pruning as a multi-objective subset-selection problem solved with evolutionary search. This is a unified design, which is applicable to CNN and Transformer

2. The evaluation metric is diverse, which makes the results and conclusion convincing.

3. The experiments use model that in both vision and language domain. The whole evaluation is thorough.

**Weaknesses:**

1. The major concern about this paper is the novelty. While the paper claims to propose the first general fairness-aware structured pruning framework, but the components in the method is a combination of well-established works. There are many references in the proposed method such as Zhou et al., 2021 and Shang et al., 2022 for subset-selection, and Deb et al., 2002 for multi-objective trade-offs, etc. Therefore, I have to ask the originality of the proposed method, because right now it seems that the method is incremental.

2. Current experiments are focusing on relatively small model and dataset. How would F-PvE scale to very large models (e.g., BERT-Large, ViT-Huge), or large dataset like ImageNet? Cooperative co-evolution may face exponential growth in search space, could gradient-based or hybrid heuristics reduce the cost?

3. F-PvE is search-based, and although the paper acknowledges higher computation costs, there’s no quantitative breakdown such as GPU-hours results.

**Questions:**

Please refer to weaknesses.

---

### Official Review · Reviewer_Xpds · 2025-11-02

**Soundness:** 3
**Presentation:** 3
**Contribution:** 3
**Rating:** 4
**Confidence:** 4

**Summary:**

The paper proposes F-PvE, a general framework for fairness-aware structured pruning. It formulates pruning as subset selection over structural units (e.g., CNN filters, Transformer heads) and uses evolutionary optimization to search for pruned subnetworks that jointly optimize accuracy and fairness. To cope with competing objectives and large combinatorial spaces, the method multi-objective evolutionary algorithms and cooperative coevolution to enhance the performance. Experiments on various datasets indicate that F-PvE achieves better or comparable accuracy while improving fairness compared to criteria-based pruning and a recent fairness-aware structured pruning baseline.

**Strengths:**

1. The problem formulation and proposed method is general to both CNN and transformers.
2. The adoption of NSGA-II for pareto-front optimization is well-motivated.
3. This paper tackles multiple fairness settings, including EDI, DED, and DT.

**Weaknesses:**

1. The core optimization machinery, multi-objective evolutionary search and cooperative coevolution, are well-established. The paper’s contribution primarily lies in instantiating these tools for fairness-aware structured pruning rather than introducing fundamentally new optimization principles.
2. Evolutionary algorithms typically require many model evaluations and fine-tuning cycles. This paper does not provide a computing comparison. This paper just say acceptable and finished within 12 hours.
3. The paper offers no theoretical analysis of when/why fairness should improve under the proposed search, nor any convergence guarantees for the fairness~accuracy Pareto optimization.

**Questions:**

NA

---

### Note · Authors · 2025-11-20

I have read and agree with the venue's withdrawal policy on behalf of myself and my co-authors.